# Relationship between Arachidonate 5-Lipoxygenase-Activating Protein Gene and Peripheral Arterial Disease in Elderly Patients Undergoing General Surgery: A Retrospective Observational Study

**DOI:** 10.3390/ijerph20021027

**Published:** 2023-01-06

**Authors:** Sejong Jin, Eun-Ji Choi, Yoon Ji Choi, Won Kee Min, Ju Yeon Park, Seung Zhoo Yoon

**Affiliations:** 1Department of Neuroscience, Korea University College of Medicine, Seoul 02841, Republic of Korea; 2Department of Dental Anesthesia and Pain Medicine, School of Dentistry, Pusan National University, Dental Research Institute, Yangsan 50612, Republic of Korea; 3Department of Anesthesiology and Pain Medicine, Korea University Ansan Hospital, Korea University College of Medicine, Seoul 02841, Republic of Korea; 4Department of Anesthesiology and Pain Medicine, Daedong Hospital, Busan 47737, Republic of Korea; 5Department of Anesthesiology and Pain Medicine, Korea University Anam Hospital, Korea University College of Medicine, Seoul 02841, Republic of Korea

**Keywords:** ALOX5AP, peripheral artery disease, single nucleotide polymorphism

## Abstract

Patients with peripheral arterial disease (PAD) are at a higher risk of developing postoperative complications. Arachidonate 5-lipoxygenase-activating protein (ALOX5AP) plays an important role in atherosclerosis pathogenesis. In this study, the relationship between PAD and several single nucleotide polymorphisms (SNPs) of ALOX5AP (rs17216473, rs10507391, rs4769874, rs9551963, rs17222814, and rs7222842) was investigated in elderly patients undergoing general surgery. The medical records of 129 patients aged > 55 years who underwent elective general surgery between May 2018 and August 2019 were retrospectively reviewed. The A/A in rs17216473, A/A in rs10507391, G/G in rs4769874, and A/A in rs9551963 were calculated as 0 points and the rest as 1 point to define the genetic risk score. The prevalence of PAD tended to increase with higher genetic risk scores (patients had less ALOX5AP gene polymorphism of A/A in rs17216473, A/A in rs10507391, G/G in rs4769874, or A/A in rs9551963) (*p* = 0.005). Multivariate logistic regression analysis revealed that the genetic risk score (*p* = 0.009) and age (*p* = 0.007) were positively correlated with the prevalence of PAD. Genetic polymorphisms of ALOX5AP and age were associated with the prevalence of PAD in this study.

## 1. Introduction

Postoperative complications increase patient morbidity and mortality. Various efforts are being made to reduce complications in the physical, mental, and emotional health of patients. Patients with underlying diseases, especially peripheral arterial disease (PAD), are at higher risk of developing postoperative complications [1].

Peripheral arterial disease is commonly known as an arterial occlusive disease affecting the lower extremities. Age, female sex, smoking, obesity, hypertension, and diabetes mellitus (DM) are well-known risk factors for PAD [2,3]. PAD is associated with coronary artery disease, or ischemic stroke, owing to its shared pathogenesis [4] and has a higher all-cause mortality rate [5,6]. 

The ankle-brachial index (ABI) is typically used as an initial tool for diagnosing PAD [7]. ABI is defined as the ratio of systolic blood pressure at the ankle to that at the arm [8]. It is a noninvasive, inexpensive, quantitative, and highly reproducible test. It can detect asymptomatic and symptomatic peripheral arterial disease (PAD). An ABI of less than 0.90 is commonly considered the threshold for PAD diagnosis. A lower ABI was found to correlate with higher all-cause and cardiovascular mortality [9].

Arachidonate 5-lipoxygenase-activating protein (ALOX5AP), also known as 5-lipoxygenase-activating protein (FLAP), is a key regulator of the leukotriene signaling pathway. ALOX5AP forms a complex with 5-lipoxygenase and activates it. The complex converts arachidonic acid to 5-HEPE, a precursor of various leukotrienes [10,11,12]. The leukotriene signaling pathway plays a crucial role in inflammation and atherosclerosis [13]. Coronary artery disease (CAD) and ischemic stroke are secondary to atherosclerosis. Therefore, various polymorphisms of the ALOX5AP gene are associated with atherosclerosis [14], coronary artery disease [15,16], and ischemic stroke [17,18]. 

Although PAD has a clinical and pathophysiological correlation with CAD and ischemic stroke, no study has examined the link between genetic polymorphisms of the ALOX5AP gene and PAD. In this study, the relationship between PAD and several single-nucleotide polymorphisms (SNP)s of ALOX5AP was investigated in elderly patients undergoing general surgery.

## 2. Materials and Methods

### 2.1. Study Population and Data Collection

This retrospective observational study was approved by the Institutional Review Board of Pusan National University Yangsan Hospital (Yangsan, Republic of Korea; approval number 05-2019-156) and registered on the Clinical Research Information Service (https://cris.nih.go.kr, accessed on 3 January 2023, KCT0007148, Registered on 4 April 2022, principal investigator: Ju Yeon Park).

We reviewed the medical records of patients who underwent elective general surgery and genetic testing between May 2018 and August 2019 at Pusan National University Yangsan Hospital. Male patients over 55 years of age with an American Society of Anesthesiologists (ASA) physical status of 1–3 were included. Patients with neurological or psychiatric disorders that could make ABI measurements inaccurate owing to poor cooperation were excluded. 

Information on age, gender, height, weight, and underlying diseases was collected from the pre-anesthetic evaluation sheet. Underlying diseases were recorded in the form of checklists for hypertension, diabetes mellitus, cardiovascular disease, respiratory disease, chronic kidney disease, cerebrovascular disease, and others, which was the result of a medical record review and pre-anesthetic interview by an anesthesiologist before surgery. ABI values and genotyping results were also collected, details of which are described below.

### 2.2. Ankle-Brachial Index (ABI)

The ABI values of the patients were retrospectively obtained from medical records. Protocolized ABI measurements were routinely performed for elderly patients undergoing general surgery just before the patients in this group received general anesthesia in the operating room at a single medical center using the standard Doppler ultrasound method [8], as described below. 

The patients lay in a supine position and rested for at least 5 min at a comfortable temperature of ~21 °C. The width of the cuff was greater than 40% of the circumference of each limb. The cuff was inflated to a pressure 20 mmHg higher than the level at which the flow signal had disappeared. The cuff was then deflated slowly. The pressure at which the flow signal reappeared was recorded as the SBP. Flo-Lab 2100-SX (Parks Medical Electronics Inc., Aloha, OR, USA) was used to detect the flow signals. The pressure in the bilateral brachial, dorsalis pedis, and posterior tibial arteries was measured in a clockwise sequence starting from the left arm. The first measurement was repeated at the end, considering white-coat hypertension. When the second value was ≥ 10 mm Hg, the first value was disregarded. Alternatively, the average of the first and second values was used. 

Ankle-brachial index was calculated by dividing the SBP of the leg by that of the arm. A larger left or right value was used as the SBP of the arm considering the possibility of subclavian artery stenosis. Larger dorsalis pedis and posterior tibial artery pressure were used as the SBP of the leg. The ABI of the left and right sides was recorded.

### 2.3. DNA Isolation and Genotype Analysis

The results of the ALOX5AP gene analysis were collected from the genotyping pool of volunteers among patients undergoing general surgery. Ten milliliters of arterial blood from each participant were used for this study. DNA was isolated using a Wizard Genomic DNA Purification Kit (Promega, Madison, WI, USA). Polymerase chain reactions were performed for amplification. Six target SNPs of ALOX5AP were identified (rs17216473, rs10507391, rs4769874, rs9551963, rs17222814, and rs7222842). Primers used (Macrogen, Seoul, Republic of Korea) are listed in Table 1. Polymorphisms in the ALOX5AP gene were analyzed by pyrosequencing using the PyroMark Q96 ID system (Qiagen, Republic of Korea).

### 2.4. Hardy–Weinberg Equilibrium

In population genetics, the Hardy–Weinberg equilibrium is the principle that the genetic composition of an entire population remains constant over generations. When a population is in Hardy–Weinberg equilibrium, it means that the whole allele frequency of the population does not change over time. This principle requires several conditions, such as random mating, no mutation or selection, and sufficiently large size of population. The *p*-value of the Hardy–Weinberg equilibrium is calculated from the difference between the observed frequency and the expected frequency using the chi-square test.

### 2.5. Genetic Risk Score

To evaluate the genetic risk for PAD by integrating each SNP of the ALOX5AP gene, a genetic risk score modified from a previous publication [19,20] was introduced. The A/A in rs17216473, A/A in rs10507391, G/G in rs4769874, and A/A in rs9551963 were calculated as 0 points; the rest was 1 point.

### 2.6. Statistical Analysis

Normally distributed continuous variables are presented as the mean ± standard deviation. Alternatively, they are shown as medians and interquartile ranges (from the first to third quartile). The Shapiro–Wilk test was used to test for the normality of distributions. Pearson’s chi-squared test or Fisher’s exact test was used for frequency comparisons. The Kruskal–Wallis rank sum test was used to compare ABI values between groups. A categorical chi-squared test was used to assess the proportional trend between the genetic score and PAD. Binary logistic regression analysis was performed to determine PAD prevalence. The genetic risk score, age, body mass index, and comorbidities (such as hypertension, diabetes mellitus, cardiovascular disease, respiratory disease, chronic kidney disease, and cerebrovascular disease) were used for the analysis. The significant variables in the univariate analysis were selected for the adjusted model.

When analyzing the differences according to SNPs, the statistical significance level was set to a *p*-value of less than 0.0125 by applying Bonferroni’s correction. Alternatively, the value was set at less than 0.05. SPSS version 26 (SPSS Inc., Chicago, IL, USA) was used for all data analysis.

## 3. Results

Data from 129 patients were analyzed. None of the patients were excluded (Figure 1). The demographic data of all patients are presented in Table 2.

The results of genotype analyses are shown in Table 3. All patients had the same genotype (G/G) for rs17222842 and rs17222814, except for one patient with A/G for rs17222814. Therefore, these two SNPs were excluded from further analyses. In this study population, the allele frequencies of the other four SNPs (rs17216473, rs10507391, rs4769874, and rs9551963) were within Hardy–Weinberg’s equilibrium.

The prevalence of PAD for each SNP is shown in Table 4. The prevalence of PAD appeared to have a sequential trend according to the genotype of each SNP, but the difference was not statistically significant. The relationship between the genetic risk score and the prevalence of PAD was then examined (Table 5). The A/A in rs17216473, A/A in rs10507391, G/G in rs4769874, and A/A in rs9551963 were calculated as 0 points, and the rest as 1. As the genetic risk score increased, the prevalence of PAD tended to increase (score 0, 25.0%; score 1, 33.3%; score 2, 65.2%; score 3, 76.1%; and score 4, 75.0%) (*p* = 0.005).

Logistic regression analysis was performed to observe the influence of genetic risk score and demographic factors on PAD prevalence (Table 6). The genetic risk score, age, body mass index, and DM were statistically significant in the univariate analysis. Therefore, these factors were included in the adjusted model. Multivariate logistic regression analysis showed that genetic risk score (*p* = 0.009) and age (*p* = 0.007) were positively correlated with the prevalence of PAD. The odds ratio of the genetic risk score was 2.153 (95% confidence interval: 1.241–3.952). DM presented borderline significance (*p* = 0.079); body mass index was not statistically significant (*p* = 0.280) in this adjusted model.

## 4. Discussion

Our primary goal was to investigate the relationship between SNPs of the ALOX5AP gene and PAD. This study showed that the prevalence of PAD is associated with genetic polymorphisms in the ALOX5AP gene. In this study, individuals with some SNP combinations of the ALOX5AP gene (except A/A in rs17216473, A/A in rs10507391, G/G in rs4769874, and A/A in rs9551963) were more susceptible to PAD.

Peripheral arterial disease is primarily caused by atherosclerosis of the lower extremities. There are several risk factors for PAD, including age, sex, smoking, obesity, hypertension, and DM. According to a multivariable analysis of a systematic review, the risk of PAD increases 1.55-fold every 10 years of age. In addition, the odds ratio of PAD was 0.74 with women as the reference [3]. Diabetes mellitus (DM) is an important risk factor for PAD. DM is strongly associated with PAD, with reported odds ratios ranging from 1.89 to 4.05 [21]. In addition, DM increases amputation and mortality rates [22]. In our study, age was found to be an independent risk factor for PAD; however, other potential risk factors were not.

ALOX5AP is one of the most important factors that regulate the leukotriene pathway. Leukotrienes are pro-inflammatory mediators implicated in inflammatory diseases, such as asthma and atherosclerosis. Leukotriene receptor modulators are already being used or studied as therapeutic targets for these diseases [23].

Atherosclerosis is a major cause of coronary artery disease and ischemic stroke. Therefore, many studies have investigated the association between ALOX5AP and these diseases. Some studies have shown an association, but others have not. In a study of a Northern Han Chinese population, the A/A genotype of rs9551963 was associated with a lower risk of atherosclerotic cerebral infarction than the others in the smoking group [24]. Another study also showed that women with the C allele of rs9551963 had a higher stroke rate [25]. Similar results were reported in a study on myocardial infarction and the ALOX5AP pathway [26]. A meta-analysis of the European population suggested that the A allele at rs10507391 is a protective factor against ischemic stroke [27]. However, in a meta-analysis of a Chinese population, the A allele was found to be a risk factor for ischemic stroke [28]. These differences are probably due to the differences in the genetic predisposition of each race. In this study, the prevalence of PAD was higher in patients with the T allele at rs10507391; however, the difference was not statistically significant. This study suggests that the prevalence of PAD increases with fewer genotypes (A/A in rs17216473, A/A in rs10507391, G/G in rs4769874, and A/A in rs9551963). By identifying the genetic risk factors, high-risk patients may have the opportunity to control for other environmental factors. Therefore, the development of this disease can be prevented.

Ankle-brachial index testing is a useful method for detecting PAD. PAD is classified into three levels of disease severity according to ABI value: mild (0.7–0.9), moderate (0.4–0.7), and severe (<0.4) [29]. A meta-analysis reported that the ABI test had a sensitivity of 61% and a specificity of 92% for the PAD diagnosis [30]. There is consensus regarding the importance of screening for PAD using ABI in patients with risk factors [31].

However, this study had some limitations. Only men were included in this study because PAD and ABI values showed different sex-specific trends [8]. Only age was found to be a significant risk factor among the factors generally considered risk factors, which may be the result of excluding the ASA class 4 patient group. In addition, this study was retrospective; PAD-related symptoms and radiological information were not included. Therefore, well-designed prospective, large-scale cohort studies are essential to clarify the relationship between genetic polymorphisms and PAD. Nevertheless, this study is valuable as an attempt to elucidate the relationship between ALOX5AP SNPs and PAD.

## 5. Conclusions

In conclusion, SNPs of ALOX5AP (except A/A in rs17216473, A/A in rs10507391, G/G in rs4769874, and A/A in rs9551963) were associated with the prevalence of PAD in this study. Some genotype combinations of ALOX5AP could be a risk factor for PAD.

## Figures and Tables

**Figure 1 ijerph-20-01027-f001:**
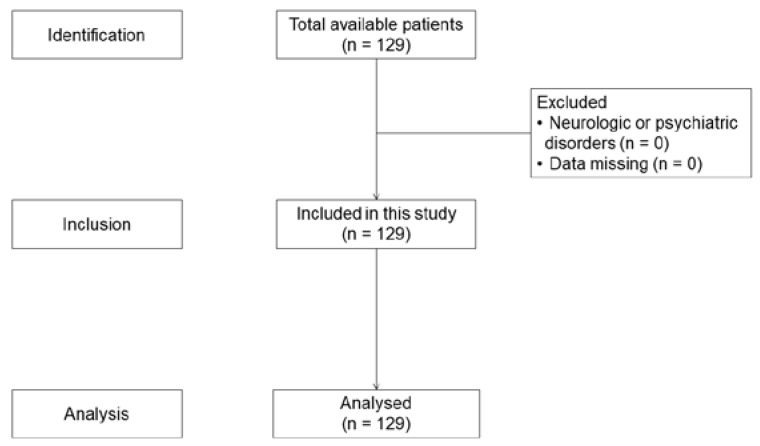
Flow diagram of the participants in this study.

**Table 1 ijerph-20-01027-t001:** Primers used for PCR amplification and pyrosequencing of regions of the human *ALOX5AP* gene.

SNP	Design Strand	Context Sequence
rs17216473	Forward	TGACCTCAGGTGATCTGCCTGCCTC[A/G] GCCTCCCACAGTTTTGTGATTATAG
rs10507391	Forward	AATAACTGTATCACTGGTGCGGGCT[A/G] TAGACATCAGCTGGGAATGAAGGTG
rs4769874	Forward	CCGTTGCGTTCTGCTCCGTCGGCCC[C/G] GAGCTGCATGGCCAACTCCCAGCAG
rs9551963	Forward	CTCACCCCGCCGCCGCCGCCGTCCC[C/G] GAGCTCCGCACAGTGTGCCCCAGCC
rs17222814	Forward	CTAGTCTCTTTCCCCAGCCACTGTT[A/G] CCCAGTGGGCTTACATATATCATGG
rs17222842	Forward	AGTTTTCCTGGGATGTGGTCCTTTC[A/G] GTTTTTTAAAAATTATTTTTATTGA

SNP, single nucleotide polymorphism. PCR, polymerase chain reaction.

**Table 2 ijerph-20-01027-t002:** Characteristics of the patients.

Overall (*n* = 129)	
Age	71.25 ± 7.15
Height (cm)	166.03 ± 9.25
Body weight (kg)	65.9 (60.2–70.7)
Body mass index (kg/m^2^)	23.94 ± 7.15
Comorbidities	
Hypertension	92 (71.32)
Diabetes mellitus	52 (40.31)
Cardiovascular disease	43 (33.33)
Respiratory disease	17 (13.18)
Chronic kidney disease	13 (10.08)
Cerebrovascular disease	20 (15.50)
Others	6 (4.65)
Prevalence of PAD	85 (65.89)

Values are shown as the mean ± standard deviation, median (quartile), or number (%). PAD: peripheral artery disease.

**Table 3 ijerph-20-01027-t003:** Genotype with allele frequencies of *ALOX5AP* gene polymorphisms.

SNP	Genotype	Frequency	Allele	Frequency	*p*-Value of Hardy–Weinberg Equilibrium
rs17216473	A/A	5 (3.9)	A	55 (21.3)	0.651
	A/G	45 (34.9)	G	203 (78.7)	
	G/G	79 (61.2)			
rs10507391	A/A	23 (17.8)	A	111 (43.0)	0.753
	A/T	65 (50.4)	T	147 (57.0)	
	T/T	41 (31.8)			
rs4769874	A/G	7 (5.4)	A	7 (2.7)	0.751
	G/G	122 (94.6)	G	251 (97.3)	
rs9551963	A/A	71 (55.0)	A	186 (72.1)	0.084
	A/C	44 (34.1)	C	72 (27.9)	
	C/C	14 (10.9)			
rs17222814	A/G	1 (0.8)	A	1 (0.4)	0.965
	G/G	128 (99.2)	G	257 (99.6)	
rs17222842	G/G	129 (100)	G	258 (100.0)	-

Values are described as numbers (%). SNP, single nucleotide polymorphism.

**Table 4 ijerph-20-01027-t004:** Prevalence of PAD for each SNP.

SNP	Genotype	Prevalence	*p*-Value
rs17216473	A/A	1 (20.0)	0.111
	A/G	31 (68.9)	
	G/G	53 (67.1)	
rs10507391	A/A	13 (56.5)	0.578
	A/T	44 (67.7)	
	T/T	28 (68.3)	
rs4769874	A/G	6 (85.7)	0.421
	G/G	79 (64.8)	
rs9551963	A/A	41 (57.7)	0.026
	A/C	31 (70.5)	
	C/C	13 (92.9)	

Values are presented as numbers (%). PAD, peripheral artery disease; SNP, single-nucleotide polymorphism.

**Table 5 ijerph-20-01027-t005:** PAD prevalence according to the genetic risk score.

Genetic Risk Score	Prevalence	*p*-Value
0	1 (25.0)	0.005
1	3 (33.3)	
2	43 (65.2)	
3	35 (76.1)	
4	3 (75.0)	

Values are presented as numbers (%). PAD, peripheral artery disease. The A/A in rs17216473, A/A in rs10507391, G/G in rs4769874, and A/A in rs9551963 were calculated as 0 points, and the rest as 1.

**Table 6 ijerph-20-01027-t006:** Logistic regression analysis for PAD prevalence.

Univariate Analysis		
Variables	OR (95% CI)	*p*-Value
Genetic risk score	2.024 (1.234, 3.4940)	0.007
Age	1.086 (1.029, 1.151)	0.004
Body mass index	0.873 (0.762, 0.992)	0.042
Hypertension	1.258 (0.560, 2.771)	0.571
Diabetes mellitus	2.796 (1.280, 6.461)	0.012
Cardiovascular disease	0.950 (0.442, 2.080)	0.896
Respiratory disease	0.533 (0.188, 1.527)	0.232
Chronic kidney disease	1.184 (0.361, 4.589)	0.789
Cerebrovascular disease	0.578 (0.219, 1.554)	0.267
**Multivariate Analysis ***		
**Variables**	**OR (95% CI)**	** *p* ** **-Value**
Genetic risk score	2.153 (1.241, 3.952)	0.009
Age	1.088 (1.025, 1.160)	0.007
Body mass index	0.924 (0.797, 1.064)	0.280
Diabetes mellitus	2.170 (0.929, 5.289)	0.079

* Statistically significant variables (*p* < 0.05) in the univariate analysis were in statistically significant variables (*p* < 0.05) in the univariate analysis were included in this adjusted model. PAD, peripheral artery disease; OR, odds ratio; CI, confidence interval. The genetic risk score was calculated as A/A for rs17216473, A/A for rs10507391, G/G for rs4769874, and A/A for rs9551963 as 0 points, and the rest as 1.

## Data Availability

Not applicable.

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
