# Peer review of "Relationship between Arachidonate 5-Lipoxygenase-Activating Protein Gene and Peripheral Arterial Disease in Elderly Patients Undergoing General Surgery: A Retrospective Observational Study"

_ijerph, 2023, doi:10.3390/ijerph20021027_

Round 1

Reviewer 1 Report

This article treats an actual problem (genetic risk of PAD) and provides about evidence that genetic polymorphisms are associated with prevalence of PAD.   Could authors provide genotyping results and DNA sequencing chromatogram? 

Author Response

Dear editors and reviewers

Thank you for your letter and for the reviewers’ comments concerning our manuscript entitled " Relationship between arachidonate 5-lipoxygenase-activating protein gene and peripheral arterial disease in elderly patients undergoing general surgery: A retrospective observational study" (ijerph-2121366). All your comments are very helpful for revising and improving our article. We made corrections one by one according to the comments of the reviewers. Here are the responses to the reviewers’ comments as follows.

<Reviewer 1>

This article treats an actual problem (genetic risk of PAD) and provides about evidence that genetic polymorphisms are associated with prevalence of PAD.   Could authors provide genotyping results and DNA sequencing chromatogram? 

: Author reply: Thank you for your comments. Unfortunately, we are unable to accept your request. The DNA sequencing chromatogram was not performed in our study group. As this study is retrospective, there is no consent for data disclosure, and even so, additional analysis is nearly impossible. When planning future prospective studies, we will consider the DNA sequencing chromatogram testing and consent for data disclosure.

Once again, thank you very much for your comments and suggestions.

Sincerely,

Yoon Ji Choi, MD, PhD

Reviewer 2 Report

Thank you for the invitation to review this good paper. overall this is a good paper with correct design and analysis. There is several minor queries need to be addressed before reviewer can accept this paper.

1. in table 2, there are confounders included in this study but no information in the method section. Please add one new section in method explaining about confounders, how do you acquire the data? is it through medical records? ICD X or IX? what is the definition of each confounders?

2. in table 3, authors mentioned about Hardy-Weinberg equilibrium and heterozygosty test but there is no explanation about that in the method section. please add information about this test.

3. regarding prevalence of ABI, authors used the medical reports data to calculate ABI but there is no explanation regarding the time. Authors used medical record data to calculate ABI at what time? before genetic test or before study? how many participants had ABI prior to study?

Author Response

Dear editors and reviewers

Thank you for your letter and for the reviewers’ comments concerning our manuscript entitled " Relationship between arachidonate 5-lipoxygenase-activating protein gene and peripheral arterial disease in elderly patients undergoing general surgery: A retrospective observational study" (ijerph-2121366). All your comments are very helpful for revising and improving our article. We made corrections one by one according to the comments of the reviewers. Here are the responses to the reviewers’ comments as follows.

<Reviewer 2>

Thank you for the invitation to review this good paper. overall this is a good paper with correct design and analysis. There is several minor queries need to be addressed before reviewer can accept this paper.

1. in table 2, there are confounders included in this study but no information in the method section. Please add one new section in method explaining about confounders, how do you acquire the data? is it through medical records? ICD X or IX? what is the definition of each confounders?

=> Author reply: Thank you for your comments. We used a pre-anesthetic evaluation sheet to collect data, and the presence of underlying disease is determined by the judgment of the anesthesiologist who performed the pre-anesthetic evaluation. We described this in section ‘2.1. Study population and data collection’ as below.

Information on age, gender, height, weight, and underlying diseases was collected from the pre-anesthetic evaluation sheet. Underlying diseases were recorded in the form of checklists for hypertension, diabetes mellitus, cardiovascular disease, respiratory disease, chronic kidney disease, cerebrovascular disease, and others, which was the result of a medical record review and pre-anesthetic interview by an anesthesiologist before surgery.

2. in table 3, authors mentioned about Hardy-Weinberg equilibrium and heterozygosty test but there is no explanation about that in the method section. please add information about this test.

: Author reply: We added information about Hardy-Weinberg equilibrium in section ‘2.4 Hardy-Weinberg equilibrium’ as below. We used the term ‘heterozygosity’ as the expected ratio of heterozygotes. However, it was deleted because it seems to cause misunderstanding and provide unnecessary information.

In population genetics, the Hardy–Weinberg equilibrium is the principle that the genetic composition of an entire population remains constant over generations. When a population is in Hardy–Weinberg equilibrium, it means that the whole allele frequency of the population does not change over time. This principle requires several conditions, such as random mating, no mutation or selection, and sufficiently large size of population. The p-value of the Hardy–Weinberg equilibrium is calculated from the difference between the observed frequency and the expected frequency.

3. regarding prevalence of ABI, authors used the medical reports data to calculate ABI but there is no explanation regarding the time. Authors used medical record data to calculate ABI at what time? before genetic test or before study? how many participants had ABI prior to study?

: Author reply: ABI tests were performed routinely for elderly patients undergoing general surgery. Tests were done in the operating room just before the patients received general. We described additional information in section ‘2.2 Ankle-brachial index (ABI)’ as below.

Protocolized ABI measurements were routinely performed for elderly patients undergoing general surgery just before the patients in this group received general anesthesia in the operating room at a single medical center using the standard Doppler ultrasound method [8], as described below.

Once again, thank you very much for your comments and suggestions.

Sincerely,

Yoon Ji Choi, MD, PhD